# Daily and Weekly Variation in Children’s Physical Activity in Norway: A Cross-Sectional Study of The Health Oriented Pedagogical Project (HOPP)

**DOI:** 10.3390/sports8110150

**Published:** 2020-11-20

**Authors:** Iana Kharlova, Maren Valand Fredriksen, Asgeir Mamen, Per Morten Fredriksen

**Affiliations:** School of Health Sciences, Kristiania University College, 0152 Oslo, Norway; iana.kharlova@kristiania.no (I.K.); marenvaland.fredriksen@kristiania.no (M.V.F.); asgeir.mamen@kristiania.no (A.M.)

**Keywords:** children, physical activity, sedentary time, accelerometry, actigraph, MVPA, HOPP

## Abstract

Background The purpose of the study was to examine differences in objectively measured moderate-to-vigorous physical activity (MVPA min/day) and sedentary time (SED min/day) between different time domains as school hours, leisure time, and weekends. An additional objective addressed causal association between body mass and MVPA. Methods. The study sample consisted of 2015 subjects (51% girls) aged 6–12 years (9.46 ± 1.76) from the Health Oriented Pedagogical Project (HOPP) in south-east Norway. Six days of MVPA min/day and SED min/day were measured using accelerometers and presented as daily averages. The differences in physical activity (PA) were investigated between the time domains of school-hours, leisure time and weekends by age and sex. Data were analyzed using one-and two-way ANOVA. Results. The relative contribution of the different time domains in overall PA was found. Daily average of MVPA min/day and SED min/day differed significantly across the three time domains. The average weekend SED was 56 ± 3.45 and 82 ± 4.12 min/day less when compared with school hours and leisure time, respectively. On average children spent 27 ± 2.74 min/day less in MVPA during school hours, compared with leisure time (*p* < 0.001), and spent by 38 ± 2.10 min/day more during weekends compared to school hours (*p* < 0.001). Boys were more physically active than girls, and less time was spent in MVPA with age. Conclusion. With the objective of increasing PA in a child population, the findings indicate that PA intervention programs should target children with higher body mass, girls more than boys, older children more than younger, and during school hours and leisure time more than on weekends.

## 1. Introduction

Physical activity (PA) improves joints, bone mass, muscular strength, cardiovascular fitness, and mental well-being, and therefore serves as a crucial component in children’s overall health [1,2,3,4]. Interestingly, findings in Norway reveal that most young children (6–12 years) participate in PA longer than the recommended 60 min of daily moderate-to-vigorous PA (MVPA) [5]. Nevertheless, a tendency of increasing body mass index (BMI) and waist-to-height ratio (WHtR) has been observed, and children weigh more today than 20–30 years ago [6]. Additionally, it has been shown that the level of physical activity decreases with age, especially when children reach puberty [7,8,9,10]. The reason may be more focus and demands on school work with increasing age and more focus on establishing social relations.

A substantial number of children are reported to a have high level of sedentary behavior (SED) [11], and SED is considered to be a major cause of non-communicable illness worldwide in all age groups [1]. The World Health Organization revealed that an increase in SED leads to various diseases as high blood pressure, elevated cholesterol levels, obesity, cardiovascular disease, and diabetes [1]. Studies have shown that treatment of overweight issues in children are challenging and have limited effect [12,13,14]. Preventing obesity and other non-communicable illness is therefore of utmost importance as these diseases have a tendency to be carried over into adulthood [13,14].

To counteract (SED) and thereby prevent non-communicable illness, monitoring of both PA and SED in children is essential to find periods in which SED may be abolished. This implies finding daily and weekly variations in PA. An optimization of PA during weekdays and weekends may be important, especially when children increase in age and tend to be more inactive. Studies have shown that reduced PA during school hours might be associated with cultural differences between countries [15,16,17]. For this purpose, each country should record their children’s habitual PA in order to monitor and prevent unwanted SED.

As WHO defines SED as a major cause of non-communicable diseases, a diagnostic of differences in PA between age groups may also be important. Defining health-related behavior, based on the amount of PA, should perhaps be used in concert with other determinants, such as overweight and obesity. In the light of this, finding an optimal time of day when activity level should be increased may make the PA easier to plan and, hence, easier to implement and to be accepted by the children.

The aim of this study is to explore PA and SED during the time domains of school hours, leisure time, and in weekends along with age differences in PA. In addition, the study aimed to consider causal association between body mass and PA.

## 2. Materials and Methods

### 2.1. Subjects and Project Study Sample

The Health Oriented Pedagogical Project (HOPP) is a seven-year longitudinal (2015–2021) large-scale cohort study of nine elementary schools from the south-eastern part of Norway. The aim of HOPP is to investigate the effect of active learning in seven schools, using two schools as controls. Active learning is a pedagogical concept of moderate to high intensity PA during lectures, either in the classrooms, inside school buildings, in gymnasium or outdoor. In addition to anticipating improved learning skills during PA, health related benefits are expected and investigated. The recruitment process and study protocol is thoroughly described elsewhere [5].

A total population of 2817 children, 6–12 years old, parents of 2297 children (82%) signed an informed consent form and agreed participation of their children. There is no information available for the 520 non-compliant children due to ethical restrictions for contact after the initial response. Overall, 2123 measurements of PA were completed. PA was measured using Actigraph wGT3X-BT (ActiGraph LLC, Pensacola, FL, USA) in all participating children. Baseline data was obtained in spring/fall 2015 (January–June and August–September), which the current analysis is based on. Not all measurements were successfully collected, either due to broken or lost accelerometers, failure to download data, and children who were absent at initial test day. The latter was due to children being ill at home, visiting the school nurse or dentist, or refusal to participate despite consent given by parents. The final sample consisted of 2015 children, comprising *n* = 1020 girls and *n* = 995 boys, whose anthropometric measurements and descriptive characteristics are presented in Table 1 and Table 2, with age and sex specific values of BMI from Skår et al. [18]. Approximately *n* = 108 children missed one or more of the anthropometric and/or descriptive characteristics, despite complete PA data. A continuous age variable was categorized as 6.00–6.44 ≈ 6, 6.45–6.99 ≈ 7 and so forth.

### 2.2. Socioeconomic Status

In Norway, parental education level is a variable that can be representative of children’s socioeconomic status (SES). The division of SES from Statistics Norway was used. In the present study, *n* = 1613 parents responded, and parents reported the highest education level of the mother/father. In all, *n* = 13 have an elementary/secondary school level of education, *n* = 298 have graduated high school, *n* = 680 have a bachelor’s degree, and *n* = 622 have a master’s degree or higher degree. No information is available for parents who did not respond.

### 2.3. Anthropometry

During spring 2015 all children completed baseline tests. At each school, the children were assembled in a classroom, half a class at a time. They underwent measurement of physiological, psychological, and anthropometric variables, although not all of them were considered in the present analysis. Height was measured to the nearest 0.5 cm, without shoes, using a SECA 213 stadiometer (SECA GmbH, Hamburg, Germany). Body mass was measured barefooted, in light clothing, using a Bioelectrical impedance analysis (BIA) scale, Tanita MC-980MA (Tanita Corporation, Tokyo, Japan). The weight of the clothing was subtracted by deducting 0.4 kg from total weight, as a standard procedure in our laboratory [6]. Other variables, such as handgrip strength, blood pressure, executive functions, and quality of life were measured, but are not applicable to the present paper.

### 2.4. Physical Activity Assessment and Data Collection

To measure PA, ActiGraph wGT3X-BT accelerometers with sampling frequency of 100 Hz at 10 s epoch (ActiGraph LLC, Pensacola, FL, USA) were used. Detailed instructions regarding how to wear and utilize accelerometers were given to children. The devices were worn for seven consecutive days on the right hip attached with an elastic band during all hours. The devices were taken off when showering or swimming, injured or ill, or when absent from school. Raw accelerometer data at 100 Hz (10 s epochs) were collected as the magnitude of the vectors (axis1, axis2, axis3). The Troiano algorithm in ActiLife 6 was then utilized, assessing 60 consecutive min of zero counts and a tolerance of 2 min of activity [19]. Minimum required time period of 8 h per day (06:00–23:59) was necessary for data analysis. Non-wear time data were excluded prior to analyses. Mean counts per min (cpm) were used to divide PA levels on sedentary (0–99 cpm), light (100–1999 cpm), moderate (2000–4999 cpm), and vigorous (≥5000 cpm) [20,21,22,23]. By summing up minutes of moderate and vigorous intensities of PA, moderate-to-vigorous PA was calculated. Sedentary time (SED) is the sum of minutes per day in that time domain (min/day). Six days of measured PA were used in the current analysis to exclude incomplete data corresponding to the first and last days when the devices were given to children and delivered to the research assistant. Then, the 6-day PA data were divided into three time domains: school hours (0800–1500), leisure time (1600–2100), and weekend time (Saturday and Sunday 0800–2100). After that, mean SED and MVPA-intensities were calculated during these time domains.

### 2.5. Data Analysis

Statistical analyses were performed with R software Version 3.5.1 (R Foundation for Statistical Computing, Vienna, Austria) and NCSS 2019 (Number Cruncher Statistical System, LLC, Kaysville, UT, USA). The figure was made with SigmaPlot 14 (Systat Inc, San Jose, CA, USA). One-and two-way ANOVA were used to explore differences between the time domains by sex and age. In the two-way ANOVA, age was dichotomized: children younger than 10 years, and 10 years and older (Table 3).

ANCOVA was used with age as a continuous variable to investigate the effect of age on PA. Tukey’s HSD (Honestly Significant Difference) test was used as a post-hoc test. Differences in PA between the sexes were analyzed by means of two sample Welch’s *t*-tests. Presence of outliers, skewness, and normality assumptions were examined visually using graphical methods prior to fitting ANOVA. Homogeneity of variance assumption was tested with residual plots which showed a similar spread of residuals against group means. In this sample, no outliers and skewness were detected. Additionally, normal probability plot followed a straight line. Thus, the data were considered to be suitable for the ANOVA *F* test.

A Pearson correlation product analyses was used to measure the association between weight (body mass) and age. Being the most important physiological factor affecting the accelerometer data, the body mass variable was used in the analyses of the variation in MVPA. Body mass was divided into age-specific quartiles (Q1–Q4). ANOVA was used to calculate difference between body mass/age and to estimate associations with the accelerometer data.

## 3. Results

### 3.1. Sedentary Behavior

A significant difference was found overall between sexes, with boys showing lower SED in all three domains. There was a significant effect of time domains on SED for the three time domains *F* (2, 6042) = 10341, *p* < 0.001. In detail, post hoc comparisons using Tukey’s HSD test showed that children spend on average 26 min/day more in SED during leisure time compared to school hours (*p* < 0.001). A significant difference was also found between weekends and school hours (*p* < 0.001), and leisure time (*p* < 0.001) with children spending 56 and 82 min/day less in SED during weekends compared with school hours and leisure time, respectively (Figure 1).

At school hours girls added 29 min/day and boys 33 min/day of SED time from 6 to 12 years. During school hours, the time spent in SED across age was added from ~120 min/day to ~150 min/day. On average, girls were 4 min/day more sedentary during leisure time across age, while boys changed SED-time by 9.5 min/day from 6 to 12 years (Figure 1).

The results suggest that SED differed considerably during weekends and differed for sex according to the Welch’s *t*-test, *t* (1981.40) = 2.30, *p* = 0.02. On average, boys were 1 min/day less sedentary than girls, 95% CI = 0.17, 2.23. This small difference has probably occurred by chance due to a large sample size and have no practical or clinical impact. Overall, in weekends girls and boys from 6 to 12 years were 6 and 14 min/day more sedentary, respectively. Interestingly, weekend SED showed a “U-shape” curve, with the highest registered SED (around 170 min/day) occurring between the ages of 8–10 (Figure 1).

### 3.2. Moderate-to-Vigorous Physical Activity

As depicted in Figure 1, MVPA showed a downward trend across age groups for all time domains, with the highest activity observed during weekends. At 6 years, the average activity was around 80 min/day, dropping overall to 45 min/day at the age of 12, representing a drop from 6 to 12 years by 4 min/day per year for girls and 6 min/day per year for boys respectively. Boys had the highest MVPA at all ages, probably due to higher PA on the weekends as shown in Figure 1.

When comparing of the effects of school-, leisure-, and weekend-time domains on PA a significant effect of the time domains was displayed, *F* (2, 6042) = 3488.8, *p* < 0.001. The lowest level of MVPA was found during school hours (Figure 1). Post hoc Tukey’s test displayed that mean leisure time MVPA was significantly different from mean school-hour MVPA. Children were 27 min/day more active during leisure time than during school-hours (p < 0.001). Additionally, they were about 11 min/day less active during leisure time compared with weekend time (*p* < 0.001). By on average 38 min/day (*p* < 0.001) children were significantly more active during weekends than during school time domain. According to Welch’s *t*-test, MVPA was significantly different between sexes during weekends, *t* (1961, 3) = 9.60, *p* < 0.001. On average, girls were about 7 min/day less active than boys, 95% CI = 5.70, 8.64. Additionally, during leisure time, male activity lasted on average 2 min/day longer, *t* (1944, 3) = 2.09, *p* = 0.04, compared to that of girls 95% CI = 0.10, 3.11.

### 3.3. Age and Sex Differences in SED and MVPA

Overall, time spent in SED was added with age and time spent in MVPA was lower during both school hours and weekend time with age. There were age and sex differences in both SED and MVPA in most of the three time domains. Sedentary behavior during school hours (*p* < 0.01) and in weekends (*p* < 0.01) were significantly higher in girls than boys. During leisure time, however, no difference between sexes were found (*p* = 0.56).

Post hoc Tukey’s analysis showed that children were more sedentary during school hours compared to weekends, varying from 43 min/day more SED at school for the youngest age group, and up to 60 min/day more for the oldest age group (*p* < 0.001). This indicating that school hours pacify children despite active recesses and PA lectures. An age difference was found as children younger than 10 years were less sedentary during school hours than those of 10 years and older, *p* < 0.01. Additionally, during leisure time and in weekends the youngest age-group was less sedentary (162 and 165 min/day; 87 and 85 min/day, respectively), *p* < 0.01.

At school, boys were more active than girls (28 and 26 min/day, respectively), *p* < 0.01. The same pattern was observed during leisure time (boys 55 min/day, girls 54 min/day, *p* < 0.01) and weekends (boys 69 min/day, girls 62 min/day, *p* < 0.03).

The youngest children engaged in significantly more MVPA than older ones during school time (5 more min/day and 25 min/day, respectively). During leisure time, children <10 years had 3 min/day higher MVPA than children >10 years (56 and 53 min/day, respectively), *p* < 0.01. Additionally, during weekends, the youngest age group showed the highest MVPA (74 and 57 min/day, respectively), *p* < 0.01.

The relative contribution of school hours on total 6-day PA of both 6 and 7-year-olds was calculated as a proportion and equaled approximately 19%. Children aged 6 (50 min/day) and 7 (47 min/day) spend more time in MVPA during weekends than during school hours (*p* < 0.001). Similarly, children aged 8–9 were 40 min/day more in MVPA during weekends than in school hours (*p* < 0.001). The average relative contribution of school hours to the overall PA of children being 18%. The 10–12 year-olds spent on average 30 min/day more in MVPA during weekends than in school hours. The relative contribution of the school hours equaled 17%. This indicates that despite spending a substantial part of the day at school, during this time MVPA contributed only 17–19% to the overall PA during the whole week.

### 3.4. Association between Physical Activity and Body Mass

A Pearson correlation product analyses revealed that weight (body mass) and age correlated with *r* = 0.69, hence indicating a strong correlation. Body mass was divided into age-specific quartiles (Q1–Q4). There was a significant difference between quartiles of SED and MVPA during all time domains (*p* < 0.01). Children in Q1 (lower body mass) had on average 6 more min/day of MVPA during school hours compared with those in Q4 (higher body mass) in all age-specific quartiles (*p* < 0.001). On weekends, children in Q1 engaged in 18 min/day more of MVPA than those in Q4 (*p* < 0.001). SED during school hours for children in Q1 was lower by 20 min/day in comparison with children in Q4 (*p* < 0.001). Additionally, children in Q4 spent 7 more min/day being sedentary during weekends than children in Q1 (*p* < 0.001). Overall, the results indicate that children spent more time in SED with each age-specific mass quartile, and less time in MVPA.

## 4. Discussion

This cross-sectional analysis compared differences in objectively measured PA during three time domains (school hours, leisure time, and weekends) in Norwegian children aged 6–12 years. Overall, girls and boys spend almost identical time in SED during weekends and school hours. During leisure time, boys spend significantly less time in SED than girls. Children spend more time in MVPA during weekends and leisure time, while SED was considerably higher during both leisure time and school hours. Boys participated in more MVPA during all time domains compared to girls. Additionally, children spent less time in MVPA and more time in SED with age in both sexes.

### 4.1. Sedentary Activity

The current study found that children spend most time in SED during weekends and leisure time. Spending too much time in SED does not benefit a healthy development in children [1,24,25]. The findings for leisure time was expected, considering this time domain involves school-associated activities, such as homework. Increasing sedentary lifestyle in the society includes high usage of portable electronic devices, such as computers and television, which may contribute to high SED during leisure time [17,26,27]. This finding supports results from a U.S. study [26], but contradicts other research. One European study covering four countries showed large discrepancies between school hours and leisure time [27], as did other studies [28,29]. The differences between findings may be explained by differences in culture. Norway has traditionally had a culture for a lot of PA, regardless of season, and according to Dalene et al., this is still true [20]. Other cultures, such as the Mediterranean countries, UK and USA, do show less habitual PA than in Norway [27,28,29].

### 4.2. Moderate-to-Vigorous Physical Activity

In the present study children obtained, on average, more than the recommended 60 min of PA per day [30]. Boys participated in more MVPA during all time domains compared to girls, which agrees with earlier studies [17,25,31,32,33,34]. Even though most of the children do achieve the recommended 60 min/day of PA, this recommendation is the minimum of what children should participate in to have healthy development. In addition, many children do not fulfill these requirements, hence an encouragement to enhance PA is recommended.

Less time in MVPA was observed during school hours, and accounting for only 18% of the overall MVPA. The fact that children, both girls and boys, spend more time in MVPA during weekend and leisure time compared to school hours, speaks more about the lack of activity during school hours, than the time spent in MVPA during weekends and leisure time. Children spent less time in MVPA with increasing age in both sexes. Similar PA trends in children have been reported in Norwegian studies [7,8], in concert with world-wide studies [19,30,35,36]. Additionally, the relative contribution of school hours spent in total MVPA was less with age. The obvious reason for this finding is that children who have a year more school usually take more classes, which in turn force the older children to spend more time in SED. This suggests a need for PA intervention programs among the oldest children in the school curriculum. Studies have indicated that PA can facilitate learning skills, improve health, and enhance well-being at school [3,4]. Therefore, time spent in school should provide an opportunity for children to be physically active and not only sitting passively behind a desk. This is why PA enhancing programs at school should be carefully systematized and put into practice to substantially increase the intensity and length of PA in children during school hours.

The most active period was weekend time, which accounted for 45%. This result was expected, as the weekend is the longest time period free from classes and homework, which allows children to participate more in different activities. This finding supports other European studies assessing PA levels in school children using accelerometers [27,28,37], pedometers [38], and using a device which measured body movement according to heart rate [16], as well as in the U.S with the help of pedometers [32,39]. However, time spent in MVPA by Chinese children during leisure time was considerably lower when compared with the other day’s segments [40]. This was explained by cultural differences.

Less time spent in MVPA with increasing age was observed during leisure time, starting as early as the age of 9. Similar result has been reported in longitudinal analyses in the U.S. [9,29], and again observed in a cross-sectional study using data from National Health and Nutritional Examination Survey (NHANES), U.S. [19]. In addition, Australian longitudinal research regarding PA of same-age children showed that less time was spent in MVPA during leisure time [10]. Together with the finding of more time spent in SED during leisure time domain, parental intervention programs with an emphasis on PA may be appropriate to increase awareness for parents on the usefulness of PA. For instance, the HOPP-study includes more active homework for the interventions schools [5].

An interesting finding is that not only less time in MVPA was observed during school hours. Children also had the lowest SED during the same time period. This may imply that children have a relatively even distribution between sedentary classroom activity and more active recess, active transport to and from school, in combination with organized PA classes. This does not mean that PA is enough during school hours, as it contributes less than 1/5 to the total MVPA during a week.

A comparable, but not similar outcome, was found regarding MVPA and SED on weekends, as boys and girls spent more time in MVPA and SED during weekends than during the other time domains. This may be explained by the increased amount of time during the day, but also that children are more in their natural habitat when it comes to free play. Perhaps children are prone to both more PA and sedentary activity if they are more or less free to govern their own time. Especially in the youngest age-groups, children are known to engage in high-intensity activity during free play, which they are permitted to do during weekends. However, they also have free time to participate in sedentary activity at their own will. It is perhaps the natural interval between MVPA and SED that is seen in more freely active children.

### 4.3. Body Mass

The present study revealed that children in Q4 of mass on average spent more time in SED compared with children in Q1. However, no causal relationships between quartiles of WHtR and PA were observed. This highlights the necessity of considering the mass of children in assessing PA and look beyond WHtR or BMI as a primary indictor of health. If a child with a heavy stature has PA for less than, let us say, 60 min/day, that child may spend more energy than a child of lighter stature in PA for 60 min/day. A similar pattern was revealed by Ness et al. [41].

There may be two explanations for the finding that lower PA occurs in heavier children. Firstly, children in Q4 of mass do actually move less. The reason may be that, as the body mass of a child increases, so does the amount of energy that is needed to perform PA, which may lead a child to more sedentary behavior in accordance with the minimum energy principle.

Secondly, higher body mass decreases the number of counts per minute measured by a triaxial accelerometer, hence the higher the mass, the lower the acceleration performed by the child. Other factors like mental, social, and contextual factors, may also play a major role in a child’s PA level, but that discussion lies beyond the scope of this paper.

### 4.4. Strengths and Limitations

This research concerns objective PA measurements with hip-worn triaxial accelerometers. These instruments typically provide accurate PA assessment [42]. A sufficiently large sample size was gathered and approximated as normal without the presence of outliers. Subjects had a wear time equal to seven days (with a one-day subtraction due to incomplete data elimination). This methodology leads to a more careful estimation of time spent in MVPA and SED. The intensity of PA is usually shown in cut-points which are commonly being assessed, depending on the study population. This gives rise to difficulties regarding comparability across various studies and populations [19,43,44,45]. Furthermore, the definition of the different time segments in which PA was assessed depends mostly on the regular daily life patterns of the population, as well as the needs of the study. In this analysis, the durations were selected according to the Norwegian daily school routines. The current research does not take into consideration alternative sorts of school-related PA, such as recess periods between classes, lunch breaks, and physical education classes. The measurements of PA of children during these classes and time domains may be addressed in future research. Apart from this, the design of this study does not allow inferences to be made regarding the casual relationships within the population.

## 5. Conclusions

The present study makes a contribution to understand differences in PA of 6–12 year-old Norwegian children during three major time domains; school hours, leisure time, and weekends. Overall, most children achieved the minimum recommendations of 60 min MVPA/day, and boys were physically more active than girls, and time spent in MVPA was lower with age as opposed to time in SED which became higher with age. The least amount of MVPA and SED was spent during school hours, indicating an even distribution between lectures and PA. Weekend time was the primary source of MVPA and SED, perhaps indicating a normal distribution of PA and sedentary behavior if children are left to free play. In addition, sedentariness was also high during leisure time, indicating the importance of stimulating PA during that time domain. The results revealed than children who had higher body mass were less active than those who had lower body mass. With the objective of increasing PA in a child population, the findings indicate that PA intervention programs should target children with higher body mass more than those with lower body mass, girls more than boys, older children more than younger, and during school hours and leisure time more than on weekends.

## Figures and Tables

**Figure 1 sports-08-00150-f001:**
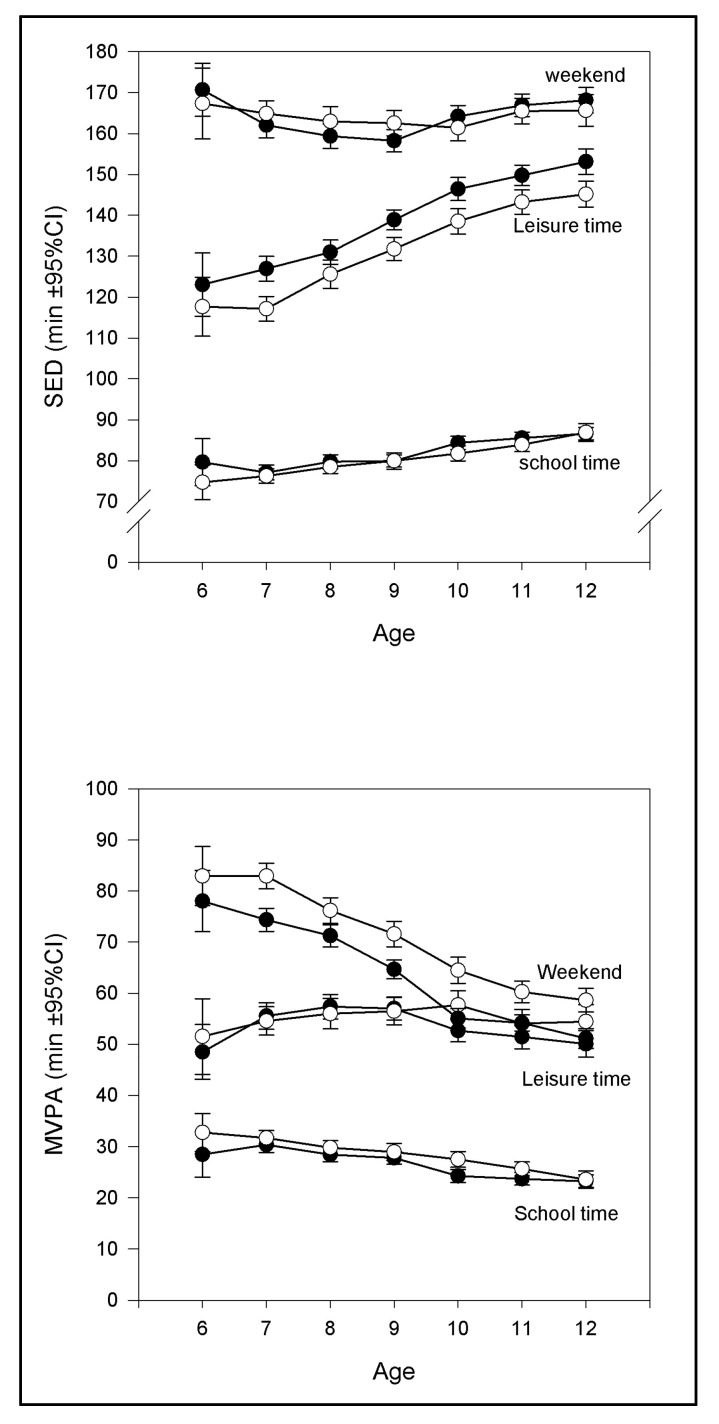
Sedentary time (SED) and moderate-to-vigorous physical activity (MVPA) during different time domains: school time, leisure time and weekends. Filled symbols represent girls.

**Table 1 sports-08-00150-t001:** Descriptive statistics of the population by age in girls.

	6 Years (*n* = 26) Mean ± SD	7 Years (*n* = 134) Mean ± SD	8 Years (*n* = 161) Mean ± SD	9 Years (*n* = 173) Mean ± SD	10 Years (*n* = 172) Mean ± SD	11 Years (*n* = 170) Mean ± SD	12 Years (*n* = 175) Mean ± SD
Anthropometry							
Mass [kg]	24.8 ± 7.4	24.3 ± 4.2	26.9 ± 4.9	31.0 ± 6.7	34.1 ± 8.2	39.4 ± 9.3	43.2 ± 9.3
Height [cm]	122.0 ± 6.1	124.1 ± 6.0	130.1 ± 6.2	135.3 ± 6.5	140.8 ± 7.1	147.3 ± 7.2	153.4 ± 8.6
BMI [kg m^−2^]	16.5 ± 3.7	15.7 ± 1.9	16.0 ± 2.2	16.8 ± 2.6	17.2 ± 3.1	17.9 ± 3.1	18.3 ± 3.2
Time spent in PA [min/day]							
SED school-hours ^†^	123.1 ± 21.1	126.9 ± 18.3	131.0 ± 19.2	138.9 ± 16.7	146.4 ± 18.2	149.8 ± 16.9	153.1 ± 19.8
SED leisure time ^†^	170.7 ± 17.6	162.1 ± 18.5	159.4 ± 19.4	158.3 ± 18.6	164.2 ± 17.3	166.9 ± 18.6	168.1 ± 20.0
SED weekend ^†^	79.6 ± 15.7	77.1 ± 11.0	79.8 ± 11.2	79.9 ± 10.3	84.4 ± 10.4	85.5 ± 10.2	86.6 ± 9.7
MVPA school-hours ^†^	28.5 ± 12.1	30.4 ± 9.1	28.4 ± 9.1	27.8 ± 8.3	24.3 ± 8.3	23.7 ± 8.2	23.2 ± 7.9
MVPA leisure time ^†^	48.5 ± 14.6	55.5 ± 15.7	57.4 ± 15.5	56.9 ± 15.3	52.6 ± 14.2	51.4 ± 15.5	50.1 ± 16.8
MVPA weekend ^†^	78.0 ± 16.3	74.4 ± 13.6	71.3 ± 14.1	64.65 ± 12.5	55.1 ± 12.3	54.2 ± 10.6	51.1 ± 12.4

SD—standard deviation; ^†^—time domains differ significantly, *p* < 0.001. SED = sedentary behavior in min/day, MVPA = moderate to vigorous physical activity in min/day, BMI = Body Mass Index. BMI values are age and sex specific based on Skår et al. [18].

**Table 2 sports-08-00150-t002:** Descriptive statistics of the population by age in boys.

	6 Years (*n* = 25) Mean ± SD	7 Years (*n* = 154) Mean ± SD	8 Years (*n* = 150) Mean ± SD	9 Years (*n* = 160) Mean ± SD	10 Years (*n* = 184) Mean ± SD	11 Years (*n* = 181) Mean ± SD	12 Years (*n* = 150) Mean ± SD
Anthropometry							
Mass [kg]	23.2 ± 4.0	24.5 ± 4.5	27.6 ± 4.5	31.3 ± 5.6	35.7 ± 7.4	39.2 ± 7.8	43.2 ± 8.7
Height [cm]	121.0 ± 5.1	125.3 ± 6.0	130.2 ± 6.4	137.2 ± 6.1	142 ± 6.6	147.4 ± 7.1	153.3 ± 7.6
BMI [kg m^−2^]	15.5 ± 1.9	15.6 ± 1.9	16.2 ± 1.8	16.7 ± 2.2	17.6 ± 2.8	18.1 ± 3.1	18.3 ± 2.7
Time spent in PA [min/day]							
SED school-hours ^†^	117.7 ± 20.0	117.1 ± 18.8	125.6 ± 21.4	131.8 ± 18.0	138.5 ± 21.5	143.3 ± 20.0	145.2 ± 19.1
SED leisure time ^†^	167.4 ± 24.1	164.9 ± 19.6	163.0 ± 22.6	162.5 ± 19.4	161.4 ± 21.8	165.5 ± 21.3	165.7 ± 23.2
SED weekend ^†^	74.8 ± 11.8	76.2 ± 11.0	78.4 ± 10.0	80.0 ± 12.5	81.8 ± 12.8	83.9 ± 11.5	86.9 ± 12.9
MVPA school-hours ^†^	32.8 ± 10.3	31.7 ± 9.0	29.8 ± 8.4	28.9 ± 10.3	27.5 ± 10.6	25.6 ± 9.4	23.6 ± 10.2
MVPA leisure time ^†^	51.5 ± 20.5	54.6 ± 17.3	56.0 ± 18.5	56.5 ± 16.7	57.7 ± 19.1	54.1 ± 18.6	54.4 ± 20.2
MVPA weekend ^†^	83.0 ± 16.0	83.0 ± 15.4	76.2 ± 15.7	71.6 ± 15.8	64.5 ± 17.7	60.2 ± 14.4	58.6 ± 14.0

SD—standard deviation; ^†^—time domains differ significantly, *p* < 0.001. SED = sedentary behavior in min/day, MVPA = moderate to vigorous physical activity in min/day, BMI = Body Mass Index. BMI values are age and sex specific based on Skår et al. [18].

**Table 3 sports-08-00150-t003:** Two-way analysis of variance in sedentary time (SED min/day) and moderate-to-vigorous physical activity (MVPA min/day). The main effect of sex and age are described in the first two lines for all time domains. A × B is the interaction between age and sex.

	F-Ratio	Prob. Level	Power (*p* = 0.05)
MVPA School Time			
A: sex	19.0	<0.001 *	0.992
B: age group	135.9	<0.001 *	1.000
A × B	0.3	0.590	0.084
MVPA leisure time			
A: sex	4.7	<0.001 *	0.581
B: age group	9.7	<0.001 *	0.874
A × B	10.4	0.001 *	0.896
MVPA weekend			
A: sex	130.8	<0.001 *	1.000
B: age group	627.3	<0.001 *	1.000
A × B	0.4	0.540	0.094
SED school time			
A: sex	78.5	<0.001 *	1.000
B: age group	408.4	<0.001 *	1.000
A × B	0.0	0.980	0.050
SED leisure time			
A: sex	0.3	0.598	0.082
B: age group	12.6	<0.001 *	0.943
A × B	10.1	<0.002 *	0.887
SED weekend			
A: sex	6.8	<0.001 *	0.738
B: age group	149.8	<0.001 *	1.000
A × B	0.2	0.657	0.073

* significant at *p* = 0.05.

## Data Availability

The datasets used and/or analysed during the current study are available from the corresponding author on reasonable request.

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
