# Peer review of "Daily and Weekly Variation in Children’s Physical Activity in Norway: A Cross-Sectional Study of The Health Oriented Pedagogical Project (HOPP)"

_sports, 2020, doi:10.3390/sports8110150_

Round 1

Reviewer 1 Report

After having verified the response to the review of the manuscript “Daily and Weekly Variation  in  Children’s  Physical  Activity.   The   Health   Oriented Pedagogical Project (HOPP)” and to the revised version of article, I would like to confirm that the current form of the manuscript almost meets the requirements of  publication in the Sports journal.

The authors have taken into account the major part of the reviewer’s comments. Although the manuscript is improved, I have some additional suggestions.

  1. In the Materials and Methods, child body mass index (BMI) should be calculated and converted to an age - and sex-specific standard deviation score based on Norwegian reference curves - the authors did not take into account the suggestions of the reviewer.
  2. Presentation of research results – I do not agree with the opinion that sex differences would not bring any additional information in this section. In Table 1 there is still no differentiation of the examined variables taking into account the sex of the examined children.
  3. I suggest in all the manuscript instead of the terms female and male, use girl and boy. After all, the group of the respondents are children aged 6-12 years.

I hope that the suggestions discussed above will be used to improve the quality of this paper.

Author Response

Referee #1

Thank you very much for your constructive comments. Here are our responses:

  1. In the Materials and Methods, child body mass index (BMI) should be calculated and converted to an age - and sex-specific standard deviation score based on Norwegian reference curves - the authors did not take into account the suggestions of the reviewer.

Response: isoBMI based on Norwegian reference material is completed.

  1. Presentation of research results – I do not agree with the opinion that sex differences would not bring any additional information in this section. In Table 1 there is still no differentiation of the examined variables taking into account the sex of the examined children.

Response: The table with the data on sex and age was updated.

  1. I suggest in all the manuscript instead of the terms female and male, use girl and boy. After all, the group of the respondents are children aged 6-12 years.

Response: Done

Thank you again for taking part in improving this manuscript.

Reviewer 2 Report

Despite all the studies showing the benefits of physical activity and exercise in general well-being, at both fitness and psychological levels, the general population seems to follow a tendency to overweight because of their inactivity. It is plausible to consider that the habit to perform physical activity begins to establish since the age period from childhood to adolescent. Thus, I appreciate the effort of the authors of conducting this cross-sectional study with such a large sample that would permit us to observe how are the habits of the young ones and, then, being able to establish practical interventions that would make the population more physically active. Although I believe that the work is well done, and for that, I congratulate the authors, below there is a list of suggestions that I consider would improve the paper's quality.

  • After reading the whole manuscript, I do not know if it would be nice to state, in the title, that the sample is from Norway. Please consider it.
  • I believe that the first part of the second paragraph should be included in the first one because the idea is very similar: the effect of the PA in general well-being. Besides, I would recommend reordering the information. First, the authors talk about the negative effects of not doing PA, then talk about positive effects, and then, again, talk about negative effects. Perhaps the solution could be beginning with the positive effects, ending with negative effects and linking with the specific topic of the study.
  • Line 65, please replace physical activity to PA.
  • Line 66, please develop a little the reason for this decrease. Do young ones choose studies over doing PA? The hours dedicated to performing works for the school increase? Are they more prone to bad habits? Maybe a whole paragraph could be dedicated to developing this idea.
  • Line 75, I found very interesting the idea "A diagnostic of differences between age groups". If it is not present in the literature, I would highlight this sentence. Or, on the contrary, if it is already present in the literature, I would report the general findings, and perhaps elaborate a paragraph with this idea.
  • Line 103, please provide the medium age and standard deviation for both female and male.
  • Table 1, please include in the notes what is SED and MVPA.
  • Line 113, just after socioeconomic status please write (SES).
  • Lines 117-119, it is not common to see the explanation of some results in the methods section. Some references are needed.
  • Lines 112-119, please consider this paragraph to be an epigraph of the methods, or subchapter just like Anthropometry. I believe that name it socioeconomic status would be suitable.
  • Line 124, just after “anthropometric variables”, please write “although not all of them were considered to analysis in this study” or something similar. Indeed, this information is already stated in lines 130-131, but, in my opinion, it should be also informed in line 124.
  • Line 129, if your laboratory has already studies which the 0.4kg deduction was applied, please add a reference. Maybe in this review process, the authors cannot provide it because of anonymity, but I suggest that in the final version these references should be included.
  • Line 151, what is SED? I found what is MVPA in line 62, but I could not find what is SED.
  • Data analysis: It would be interesting to report the effect size of the analysis. Please consider it.
  • Tables, I believe that the design of the tables should not present lines in each cell.
  • Line 290, some references are needed.
  • Line 294, please provide some references.
  • Line 296, the countries are similar to Norway? Because it would be interesting ending the paragraph with a brief explanation of why there are differences between countries and what information incorporates this study.
  • Line 359, there are not results present in the literature that supports the findings of this study? There is not any reference in the whole paragraph.
  • Line 390, I believe that is important to highlight that the sample is from Norway.

Author Response

Thank you very much for your constructive comments. Here are our responses:

After reading the whole manuscript, I do not know if it would be nice to state, in the title, that the sample is from Norway.

Response: Thank you! Very much appreciated. It is now stated.

I believe that the first part of the second paragraph should be included in the first one because the idea is very similar: the effect of the PA in general well-being. Besides, I would recommend reordering the information. First, the authors talk about the negative effects of not doing PA, then talk about positive effects, and then, again, talk about negative effects. Perhaps the solution could be beginning with the positive effects, ending with negative effects and linking with the specific topic of the study.

Response: Excellent suggestion! Done.

Line 65, please replace physical activity to PA.

Response: Thank you – done.

Line 66, please develop a little the reason for this decrease. Do young ones choose studies over doing PA? The hours dedicated to performing works for the school increase? Are they more prone to bad habits? Maybe a whole paragraph could be dedicated to developing this idea.

Response: We have elaborated on the idea. With more demands on school work, we are more prone to keep up with social contacts. Technically, with increased mass, acceleration is reduced, and there is less of a response on the accelerometer.

Line 75, I found very interesting the idea "A diagnostic of differences between age groups". If it is not present in the literature, I would highlight this sentence. Or, on the contrary, if it is already present in the literature, I would report the general findings, and perhaps elaborate a paragraph with this idea.

Response: Thank you. We have elaborated on the idea, and suggested that perhaps PA (or lack of PA) may be used as a diagnostic tool in concert with overweight and obesity.

Line 103, please provide the medium age and standard deviation for both female and male.

Response: Mean age and SD for both age and sex are now reported in table. Table 1 is now divided into table 1 and 2

Table 1, please include in the notes what is SED and MVPA.

Response: Done. Table 1 is now divided into table 1 and 2, and SED and MVPA are stated in both tables.

Line 113, just after socioeconomic status please write (SES).

Response: Done

Lines 117-119, it is not common to see the explanation of some results in the methods section. Some references are needed.

Response: Agreed. The comments are omitted.

Lines 112-119, please consider this paragraph to be an epigraph of the methods, or subchapter just like Anthropometry. I believe that name it socioeconomic status would be suitable.

Response: Agreed. Subheading added.

Line 124, just after “anthropometric variables”, please write “although not all of them were considered to analysis in this study” or something similar. Indeed, this information is already stated in lines 130-131, but, in my opinion, it should be also informed in line 124.

Response: Agreed.

Line 129, if your laboratory has already studies which the 0.4kg deduction was applied, please add a reference. Maybe in this review process, the authors cannot provide it because of anonymity, but I suggest that in the final version these references should be included.

Response: Agreed. We have cited one of our earlier papers.

Line 151, what is SED? I found what is MVPA in line 62, but I could not find what is SED.

Response: Sedentary behavior = SED. Both MVPA and SED are mentioned in the abstract. SED is also now mentioned in the methods, and added in table 3 (table 2 is now table 3).

Data analysis: It would be interesting to report the effect size of the analysis. Please consider it.

Response: At HOPP, we do consider the effect of increased PA in primary schools. However, in the present analyses, we only consider cross-sectional data from the baseline values, and not any changes due to increased physical activity. We, therefore, will include the effect size in the future analyses where we will consider the effect of intervention in this study.

Tables, I believe that the design of the tables should not present lines in each cell.

Response: The lines are removed

Line 290, some references are needed.

Response: Done

Line 294, please provide some references.

Response: Done

Line 296, the countries are similar to Norway? Because it would be interesting ending the paragraph with a brief explanation of why there are differences between countries and what information incorporates this study.

Response: The studies by Nilsson et al., Steele et al. and Wickel & Belton are not necessarily comparable with the present study. Nilsson et al. did include Norway and Denmark, but also Portugal and Spain. Spain and Portugal have quite different cultures when it comes to PA. Steele et al. investigated children in UK, and Wickel & Belton investigated children in USA, both with PA cultures quite different from that of Norway. We have tried to briefly include an explanation for the differences in the text.

Line 359, there are not results present in the literature that supports the findings of this study? There is not any reference in the whole paragraph.

Response: Thank you for pointing this out. The message in the paragraph conveys that mass has to be taken into consideration when assessing PA. We have encountered some difficulties when trying to assess papers covering this topic in children, but we found one. However, the statement about the “minimum energy principle” and, the two now added, formula of acceleration (m/s2) and the length of lever (leg length), we consider to be basic physics that need no citing as they are self-explanatory. The paragraph is rewritten, hopefully with an acceptable quality.

Line 390, I believe that is important to highlight that the sample is from Norway.

Response: Again, thank you. It is included in the conclusion.

Thank you again for taking part in improving this manuscript.

Reviewer 3 Report

The increase of hypokinetic way of life (lack of physical activity) is worldwide problem in several developed countries. There are also differences in physical activity of girls and boys, PA performed during school hours and during weekend are also known. Less known are suggestions from authors how to solve this problem. And these suggestions I expected in the end and recommendations of the study.

From the point of view of formulating the research plan (problem) and the methodology, I consider used methods as adequate.

Author Response

Referee #3

Thank you very much for your constructive comments. Here is our response:

The increase of hypokinetic way of life (lack of physical activity) is worldwide problem in several developed countries. There are also differences in physical activity of girls and boys, PA performed during school hours and during weekend are also known. Less known are suggestions from authors how to solve this problem. And these suggestions I expected in the end and recommendations of the study.

From the point of view of formulating the research plan (problem) and the methodology, I consider used methods as adequate.

Response: I believe we have covered the comments from referee #3 when responding to the comment of referee #1 and #2.

Thank you again for taking part in improving this manuscript.

Round 2

Reviewer 2 Report

Congratulations for the good work and thank you for considering my suggestions. I believe that now the paper could be accepted.

Author Response

Thank you very much for your comment - very helpful and appreciated.

This manuscript is a resubmission of an earlier submission. The following is a list of the peer review reports and author responses from that submission.

Round 1

Reviewer 1 Report

The paper is a study that analyzed an interesting topic, children’s circadian variation in physical activity.

However, there are some ambiguities that require clarification.

The aim of the study was to investigate physical activity (PA) and sedentary time (ST) during school hours, after-school hours, and on weekends along with the age differences in PA, in order to find the optimal time periods in which to start the intervention. However, this goal has not been fully achieved. The Authors did not indicate directly (specifically) the optimal time at which the intervention should be started. The only very general assumption, given in Conclusions, should be clarified. Introduction is not a good justification for formulating the mentioned purpose of the study. The whole section is not interesting and is not a good introduction to further analysis of the issue. The information included in this part of the study should not be based solely on the Authors' feelings, but primarily on specific references to the literature on the subject. The theoretical section should explain the topic as well, to better understand the phenomenon under study. In the Materials and Methods, a more detailed description of the cited longitudinal study (HOPP) is necessary. The description of the empirical material presented in this study is incomplete. There is no information on where the research was carried out, and in what environment (rural, urban). In addition, in section 2.2, an anthropometric measurement procedure should be described in more detail (description of test conditions). Part 2.3 lacks the time period for the tests. The calculation of the nutritional status of children based on the BMI index is a serious mistake. Child body mass index (BMI) should be calculated and converted to an age - and sex-specific standard deviation score based on Norwegian reference curves. Presentation of research results in the Results section was made on the basis of incomplete explanations which did not allow to syntactically explain the next stages of the research process. In Figures 1, 3, 5 and in Table 2 there is no differentiation of the examined variables taking into account the sex of the examined children. This causes incomplete recognition of the studied phenomenon. The data contained in Table 2 were not commented on in the text. In turn, the commentary on Figure 3 - "Figure 3 illustrates averaged SED and MVPA for different age groups showing that SED increases with age, while MVPA declines." - is not entirely correct. The design of the Figure 4 is not clear and legible; there is no definition/description of the term „Quartile of mass”. Also, why didn’t the Authors calculate the effect size? The conclusions need to be revised to highlight the main ideas of the study. The discussion needs to be extended to underline the relevance of the findings related to previous research studies. This section does not help to identify what the results provided in the light of existing body of knowledge that could bring the discussion forward.

Author Response

The title has been changed.

A sentence has been added in conclusion.

The introduction has been changed and the methods section has been expanded.

We agree that reference curves of BMI may give more information. However, BMI was not in the scope of the present analysis, but rather was used for descriptive purposes of the sample using only mean and SD. We hope for your understanding on this.

Comment from reviewer 1: "Results section was made on the basis of incomplete explanations which did not allow to syntactically explain the next stages of the research process." Response: As we understand, both the data analysis and results sections have been explained after the best of ours knowledge. We therefore have some difficulties in understanding this comment from reviewer 1.

The following is related to Fig 1, 3, 5 & Table 2. In Fig 1, there's no differentiation by sex because sex differences equalled only 1 minute during weekends. There were no significant differences between females and males during other time periods. Same goes for Fig 3. There were no significant sex differences in SED, and only a 2-minute difference between females and males in MVPA. As we understand, it is not required to depict such a tiny difference in a figure because it would not be visible in a bar chart, and in some cases there were no significant differences between sexes. Table 2 presents descriptive characteristics by age, not by sex. The whole section is dedicated to age differences. We believe that including sex differences would not bring any additional information in this section. Hopefully, this is acceptable.

Description of Fig 3 was slightly changed. We believe that in this context most of the readers have an understanding of the concept “quartile”, and we prefer to keep the text as it is without going into details on the definition. Hopefully, this is acceptable.

We agree that the effect size is a valuable measure. However, in this research, the three groups were used, not two, which is the traditional way of finding the effect size. If reviewer 1 agrees, we would like to keep the present analyses without including the effect size.

Comment: "The discussion needs to be extended to underline the relevance of the findings related to previous research studies. This section does not help to identify what the results provided in the light of existing body of knowledge that could bring the discussion forward." Response: We believe we have covered the essentials of the variation in physical activity. We have also, in agreement with reviewer 1, changed some parts of the text.

Reviewer 2 Report

This work is interesting and deals with a subject that represents a real current issue
Note the positive point of a large sample size (2015 subjects). However, elements of analysis and interpretation of these results are fragile and need to be reviewed:
- the different age groups have various numbers of subjects, in particular the 6 year old group is much smaller than the other groups. It is a limit to the statistical analysis.
- It is difficult to compare the PA levels of these 6-12 year olds without then discussing the very specificities relating to the needs of the children, on the one hand, but also and above all the school requirements of the children. Indeed, the older the children are, the more the constraints of school learning for these older children force them to remain inactive.
Finally, the nutritional aspects are not mentioned, even though they are decisive for studies on PA and inactivity. If children are overweight and/or obese, there are effects on their PA levels. It would therefore have been interesting in this framework to look for possible correlations between BMI and PA, and not to limit oneself to an ANOVA.

Finally this work is relevant, but it does not bring any new facts into the discussion : the results are analysed in a very 'traditional' way without trying to highlight elements that could bring a new approach at these data.
For example, cluster classification analyses could allow to highlight specificities of practice, gender, etc.

In addition, remarks are directly annotated in the attached manuscript.

Author Response

Please see the text for additional comments.

As we understand, the term "sex" refers to the biological explanation of the difference between females and males. The expression "gender", on the other hand, refers to the sociological description, and may also include people with intersex conditions. We suggest using the biological description.

Comment: "I think that the intervention must also consider the nutritional aspects…" Response: We believe that nutrition is not within the scope of this paper. Future papers will cover this topic.

Comment regarding Table 2: "I don't know what you mean here. If it's related to age groups, why in Table 2; the age groups are presented in one-year increments?" Response: We have difficulties in understanding what the reviewer is suggesting.

A response regarding processing algorithm: Raw accelerometer data at 100 Hz (10 s epochs) were collected as the magnitude of the vectors (axis1, axis2, axis3). The Troiano algorithm in ActiLife 6 was then utilized, assessing 60 consecutive mins of zero counts and a tolerance of 2 mins of activity.

Comment: "Is it possible to report the symbols of significant differences?" Response: In Fig 1, 2 & 5 it is very inconvenient.

The reviewer suggested deleting Fig 3. We would like the reviewer to reconsider this as we believe the figure gives a very nice visual representation.

A more conventional method as ANOVA was used first and foremost because it is easier to compare with different studies, as all the cited studies used this method. Your suggestion regarding using new approaches would indeed bring a contribution in this field. We will surely consider your suggestion in future research.

Reviewer 3 Report

The term “circadian” occurs only twice in the entire manuscript (title and introduction) with no references to it in the discussion section of the manuscript. Since circadian rhythms is mainly a term utilized when evaluating physiological, behavioral, or other factors during a 24-hour period, it is suggested that you delete any reference to the term “circadian” in this manuscript.  While the outcome variables are being evaluated during the course of a day for two of the time points, the separate evaluation of weekend SED and PA time seems unrelated to an evaluation of a circadian response.

Lines 43 and 50 are redundant. It is suggested that BMI percentiles be included in your data analysis as this is seen as a more appropriate reportable measure of BMI in children. It is recommended that total daily SED and MVPA be included in Tables 1 and 2. This would provide more context for amount of time spent in each of the reported time frames.

Results section 3.4 Association between physical activity and body mass – Are the associations between body mass quartile and PA a function of body mass or age? Line 199 – Revise the word “changes” to “differences” for clarity based on your study design.

Line 208-211 – The authors state that since PA improves learning, health, and well-being at school then schools should provide opportunities for physical activity and reduce sedentary time. It would be beneficial to add further evidence that it is the in-school PA that would drive improvements in learning instead of just an overall increase in daily physical activity. 

Additional information on some of the reported mechanisms (increased blood flow, altered catecholamines and other hormones, etc.) by which in-school PA improves learning would be helpful to provide support for the argument that children should do more PA during school hours and not just more PA during the day.

Line 216 – Change “awaited” to “expected”

Author Response

All comments suggested by the reviewer were taken into account. See edits in the text.

Round 2

Reviewer 1 Report

After having verified the response to the review of the manuscript “Children’s Circadian Variation in Physical Activity: A Cross-Sectional Study of the Health Oriented Pedagogical Project (HOPP)” and to the revised version of article, I would like to confirm that the current form of the manuscript doesn’t meets the requirements of  publication in the Journal Sports. The authors have not taken into account the major part of the reviewer’s comments.

The explanation provided is insufficient. The changes in the text of the manuscript do not correspond to the comments given in the review, they are superficial, often non-substantive.

Reviewer 2 Report

corrections have not been made and the responses made to the reviewer are not acceptable as they do not argue that no corrections were made.

it is not possible to publish an article with figures that do not meet a minimum of rigorous scientific presentation, such as the symbol to identify statistical differences.

In your response about the algorythm used to analyze your data, you specify the name of the algorithm used but this must be inserted in your methods because it is an important information.